# Designing a Compact Dual-Band, Dual-Polarized Antenna for Biotelemetry Communications Using the Characteristic Mode Method

**DOI:** 10.3390/s23229094

**Published:** 2023-11-10

**Authors:** Xiaoming Xu, Zhiwei Song, Yuchao Wang

**Affiliations:** School of Electrical Engineering, Xinjiang University, Urumqi 830017, China; xxm16@stu.xju.edu.cn (X.X.); 107552204406@stu.xju.edu.cn (Y.W.)

**Keywords:** biotelemetry, characteristic mode method, dual-band, dual polarized, implantable antenna

## Abstract

A compact dual-band, dual-polarized, implantable antenna was designed using the characteristic mode method for operation in the wireless medical telemetry service (WMTS) band (1.4 GHz) and the industrial, scientific, and medical (ISM) band (2.45 GHz). By utilizing slotting techniques and materials with high dielectric constants, the antenna volume was minimized to 47.4 mm^3^, and circular polarization was gained in the WMTS band by using the characteristic mode method. A three-layer physical model of human tissues was constructed in a simulation, and a biogel and minced pork were used in the measurements. The measured peak gains are −20.85 dBi (1.4 GHz) and −22.15 dBi (2.45 GHz). The measured effective axis ratio bandwidth in the WMTS band is 170 MHz (1.33–1.50 GHz, 12.0%), and the impedance bandwidth in the ISM band is 390 MHz (2.21–2.60 GHz, 16.2%). At 1.4 GHz and 2.45 GHz, the largest 1 g average SAR values are 376 W/Kg and 318 W/Kg, which comply with IEEE C95.1-1999. Moreover, when the communication chain affinity exceeds 20 dB for 1.4 GHz and 2.4 GHz, the transceiver range reaches 8.2 m and 9.7 m. This antenna can be used for implantable wireless telemetry systems.

## 1. Introduction

With the rapid growth of mobile telecommunications and electronic technology, the antennas of implantable wireless medical devices have become a popular research topic [1]. These devices can collect and transmit physiological data, such as blood glucose monitors, and pulmonary artery pressure monitors [2,3,4]. To enable these devices to communicate with external devices stably and in real time, it is necessary to develop implantable antennas that provide reliable performance.

There are stringent requirements and restrictions for human-implantable devices; therefore, designing implantable devices is a great challenge, and the devices must perform with appropriate characteristics such as miniaturization, polarization diversity, biocompatibility, and reliability [5]. In [6,7,8,9], the authors utilized slotting techniques to achieve a miniaturized design. In [6], the authors minimize the volume of an antenna by cutting two appropriate rectangular slots in the annular ring, resulting in a total size of about 120.69 mm^3^ including the substrate and superstrate. In [10], the authors chose a Rogers RO 3010 (ε_r_ = 10.2) as a dielectric substrate and used a slotting method to extend the current path, which also achieved a reduction in the antenna’s size, obtaining a size of 88.09 mm^3^. In [11,12], both the radiation patch and GND were slotted to achieve a miniaturized design. However, the main drawback of miniaturized antennas is the loss of efficiency and gain, particularly when placed in lossy peripheral tissues, such as in the human body. Therefore, to meet these demands, many engineers have suggested different techniques for realizing such trade-offs [13,14]. In addition, saving energy and extending the lifetimes of implantable medical devices are other challenges. In [15,16,17,18,19,20,21,22], scholars used the low power consumption of an antenna in a higher-frequency band to wake up the device; using a higher level of power consumption in a lower-frequency band to transmit data could extend the life of a device. In [16], a compact dual-band antenna was proposed to extend the device’s lifetime, and its volume was 103.7 mm^3^. In [18], the dual-band was achieved via unfolding a circular patch antenna, and the antenna’s dimensions were π × 10^2^ × 2.54 mm^3^. In addition, circularly polarized (CP) antennas are important for achieving high-quality telecommunications with outside devices as the human body’s posture and movement change [23,24,25,26]. In [25], the antenna’s dimensions were 11 × 11 × 1.27 mm^3^ with a bandwidth of 889–924 MHz, and its axial ratio (AR) bandwidth was only 11 MHz.

Therefore, interest in devising a compact, multi-frequency, implantable antenna with dual-polarization or multi-polarization characteristics to meet the needs of biological devices has been increasing.

In this work, a miniaturized dual-band, dual-polarized implantable antenna for biotelemetry communications was designed by using the characteristic pattern method and slotting method. Its total size is only 47.4 mm³ (7.4 mm × 7.2 mm × 0.889 mm). Its measured peak gain is −20.85 dBi at 1.4 GHz and −22.15 dBi at 2.45 GHz. Its measured effective AR bandwidth in the WMTS band is 170 MHz (1.33–1.50 GHz, 12.0%), and its measured impedance bandwidth in the ISM band is 390 MHz (2.21–2.60 GHz, 16.2%). We simulated and measured radiation patterns with quasi-omnidirectional characteristics at those two operating frequency points. The simulated maximum 1 g average SAR values are 376 W/Kg and 318 W/Kg at 1.4 GHz and 2.45 GHz, respectively. Therefore, the largest transmit power must be less than 4.26 mW and 5.03 mW to comply with the IEEE C95.1-1999 standard. Moreover, the transceiver has a transmission range of 8.2 m and 9.7 m when the transmission chain margin exceeds 20 dB at 1.4 GHz and 2.4 GHz, respectively.

To gain a more intuitive understanding of the characteristics of the antenna designed in this article, we list some main performance metrics for the antenna described in this article and compare them with those in the following references in Table 1. Firstly, the total size of the designed antenna in this article is the smallest. Secondly, only our work and the antenna in [16] have dual-polarization characteristics. Compared with [16], the peak gains, effective AR bandwidth, and SAR of our design are better, and only the impedance bandwidth in the ISM band is poor. Therefore, our design is a good candidate for use in biotelemetry wireless medical equipment applications, such as continuous wireless blood glucose monitoring and blood pressure monitoring.

## 2. Antenna Design and Optimization

### 2.1. Antenna Geometry

We assume the length and width of the rectangular radiation patch are L and W, respectively, and the radiation patch and GND are displayed in Figure 1a,b. The detailed variable settings and their corresponding values are listed in Table 2. The total size of the antenna is 47.4 mm^3^. Its multi-frequency characteristics can be obtained by cutting multiple I-grooves in the right part of the patch. To further increase the bandwidth, the thickness of the substrate can be adjusted appropriately. Its other dimensions and geometry are displayed in Figure 1c. The antenna is energized via a coaxial feed probe with a 0.2 mm radius, located at the right of the ground plane, which enables excellent impedance matching and dual-frequency behavior by adjusting the feed location appropriately. The radiation patch and GND are on the upper and lower surfaces in the dielectric substrate, respectively. This structure protects the radiation patch and prevents it from coming into contact with human tissue. The antenna was implanted in a three-layer body tissue model for simulation analysis, as shown in Figure 1d.

The three-layer body tissue model consisted of skin, fat, and muscle. The muscle box measured 80 mm × 80 mm × 32 mm, the fat box measured 80 mm × 80 mm × 4 mm and was located above the muscle box, and the skin box measured 80 mm × 80 mm × 4 mm and was located above the fat box [27]. The electrical behavior of the modeled environment was similar to that of human tissue. The dielectric constants and conductivities of these body tissues at two frequencies are listed in Table 3. For instance, the dielectric constant of skin tissue at 1.4 GHz is 39.6 and the conductivity is 1.036 s/m, but the dielectric constant of skin tissue at 2.4 GHz is 38.0 and the conductivity is 1.464 s/m [28]. Due to differences in the physical characteristics of human tissues at different operating frequencies, the radiation characteristics of implanted antennas will be affected. We assumed that the antenna was located in the middle of the tissue model and implanted at a depth of 20 mm. The S11 and AR simulation results at different frequencies are displayed in Figure 2. The effective AR bandwidth in the WMTS band and the impedance bandwidth in the ISM band cover the design goal very well, as shown in the shaded part of Figure 2a,b.

### 2.2. Evolution of Antenna Geometry

The designed antenna gained the desired design requirements in three main steps. The dual-frequency antenna in this article was obtained via the evolution of the planar inverted-F antenna (PIFA), and the initial dimensions of the PIFA were obtained via the following equation [29]:(1)$$f_{r}=\frac{c}{4\left(w+l\right)}$$
where *c* represents the velocity of light in free space, *w* and *l* represent the width and length of the radiation component, and *f_r_* represents the operating frequency.

The calculated center frequency of the antenna is around 5.1 GHz when the radiation patch is without slots, and the antenna does not gain dual-band characteristics. As shown in Table 3, the design is divided into three main steps. Step 1: By cutting two F-shaped slots into the left part of the radiation patch, the resonant frequency is reduced to around 2.4 GHz due to the increased current path, as shown in Figure 3a. Step 2: After cutting three I-slots into the right of the radiation patch, the antenna has a new resonance point near 1.4 GHz but it cannot cover the 2.4 GHz band well; see the green line in Figure 3a. Additionally, the AR in the 1.4 GHz band is poor as shown in Figure 3b (green line). Then, we used the characteristic mode method to analyze its polarization characteristics. The simulated model significance (MS), determined using CST, is shown in Figure 4a. The MS values of modes 1 and 2 are less than 0.8 in the 1.4 GHz band, and the simulated current vectors of modes 1 and 2 are not orthogonal, as shown in Figure 5a. So, its AR is not good enough in this case. The MS values of modes 3 and 4 are less than 0.2, so we do not consider these two modes. To gain CP characteristics in the 1.4 GHz band, step 3 is led out, and we adjust the feed appropriately and cut a cross slot on the GND. The simulated MS for the antenna and the current vector distribution at the patch surface at this time are displayed in Figure 4b and Figure 5b, respectively. The MS values of mode 1 and mode 2 are nearly equal to 1 in this case. Moreover, the simulated current vectors of the two modes are orthogonal to each other, allowing the antenna to operate in the desired band and achieve CP characteristics (Table 4).

From this, we can draw the following conclusion:(1)Because of the slots, the dimensions of the antenna are reduced for equal operating frequencies.(2)Two operating bands can be produced by cutting I-shaped grooves into the right side of the rectangular radiation patch.(3)The cross grooves cut on the GND allow for better tuning of the resonant frequency and CP performance.

### 2.3. Parameter Optimization

This section describes the impact that some key parameters have on antenna impedance matching and CP characteristics. The simulated S11 results with different thicknesses of the substrate (H_1_) and superstrate (H_2_) are shown in Figure 6a. The H_1_ and H_2_ dimensions affect both bandwidth and impedance matching; because their sizes are very small with respect to the wavelength, their variations affect the magnetic field distributions greatly. When H_1_ = 0.635 mm and H_2_ = 0.254 mm, the antenna performs well in both bands. The simulated S11 results with different lengths (L_1_) and widths (W_2_) of the cross-groove are shown in Figure 6b. The variations in W_2_ and L_1_ have a significant effect on higher-frequency bands, while lower-frequency bands remain nearly constant. The AR bandwidth values for different lengths (L_1_) and widths (W_2_) of the cross-groove are shown in Figure 7a. The capacitive loading formed by the cross-gap affects the current distribution at the radiation surface (as shown in Figure 5), and its parameters can be applied to adjust the circular polarization characteristics of antennas. When L_1_ = 7 mm and W_2_ = 0.6 mm, the impedance matching can cover both operating bands well, and the AR can cover the 1.4 GHz band well. The simulated AR when the cross-slot rotates at different degrees is shown in Figure 7b. The rotated degree of the cross-slot can change the center frequency of the AR. As the angle increases (0° to 45°), the center frequency shifts toward the higher-frequency band, so the best effect is achieved at 0°.

## 3. Analysis of Simulation Results

### 3.1. Working Mechanism of the Antenna

The antenna proposed in this article is an improvement on the rectangular patch antenna. Introducing slots into the patch not only increases the current routes and reduces the dimensions of the antenna, but also achieves CP characteristics.

For a further illustration of the operation of the antenna, Figure 8a displays the surface current distribution when the antenna operates at a 1.4 GHz frequency point. The current is mainly distributed in the left part of the patch, while the current in the GND is mainly concentrated near the right slot. Figure 8b reveals the current distribution when the antenna operates at the 2.45 GHz frequency point: the current is mainly concentrated in the center of the patch, while the current in the GND is concentrated near the right-hand slot. According to the planar inverted F-type antenna equations in Section 2 [25], the current path increases due to the two F-shaped slots and three I-shaped slots in the patch. Therefore, combining the current distribution plot and the equations, the new resonant frequencies of the antenna can be calculated to be about 1.41 GHz and 2.35 GHz.

### 3.2. Radiation Pattern and SAR

The simulated radiation patterns of the antenna in the three-layer body tissue model are displayed in Figure 9. The antenna peak gains are −20.85 dBi and −20.65 dBi at 1.4 GHz and 2.45 GHz and maintain good omnidirectionality at the two working frequency points, meeting the operational requirements for an implantable medical monitoring system.

The simulated SAR distribution of the designed antenna within the three-layer body tissue model is shown in Figure 10. Assuming a transmitting power of 1 W, the maximum 1 g average SAR values are 376 W/Kg and 318 W/Kg at 1.4 GHz and 2.45 GHz. Therefore, the maximum transmit power ought to be less than 4.26 mW and 5.03 mW to satisfy the requirements of the IEEE C95.1-1999 standard [30]. The maximum power allowed here is far less than the transmitter power of commercially available transmitters, so in this case, the SAR meets safety considerations for humans.

## 4. Measurement Results Analysis

A prototype was fabricated, as displayed in Figure 11a. The antenna was tested using an Agilent Vector Network Analyzer (PNA-X) and an anechoic chamber developed by Electronic Forty, as displayed in Figure 11b,c. The simulated and measured S11 results are illustrated in Figure 12a, and the simulated an measured AR results are displayed in Figure 12b. The impedance bandwidths of the two bands in the biogel are 0.37 GHz (1.15–1.52 GHz) and 0.41 GHz (2.20–2.61 GHz), respectively, and the AR at 1.4 GHz is 0.38 GHz (1.32–1.71 GHz). In the freshly chopped pork tissue, the impedance bandwidths of the two bands are 0.35 GHz (1.17–1.52 GHz) and 0.40 GHz (2.20–2.60 GHz), and the AR bandwidth is 0.4 GHz (1.33–1.73 GHz) in the 1.4 GHz band. The measured data are in good agreement with the simulation data, but there are some deviations in the high-frequency band. One cause is that although the test environment is very close to the simulation environment, they are not the same. Another reason is that the higher the frequency, the more significant the errors introduced by RF connection lines and connectors are.

Figure 13 shows the measured and simulated radiation patterns at the two operating frequency points. The measured results agree with the simulated results in that the antenna’s magnetic field has quasi-omnidirectional radiation characteristics field at the two operating frequency points.

## 5. Link Budget

To gain a more comprehensive understanding of the antenna’s data transmission capability, its link margin (LM) must be calculated. It is well known that the body is a multi-layered, lumped medium, and inside the human body is a complex electromagnetic environment. So, we simplified the external propagation link into a free-space model (unit: mm), and the formula of the path loss *L_f_* is as follows [29]:(2)$$L_{f}=20\log_{10}\left(4\pi d/\lambda \right)$$

Here, *d* represents the distance between the Tx and Rx antennas, and *λ* represents a wavelength in free space with respect to the working frequency. The *LM* is used to measure the communication behavior of the implanted antenna. The *LM* can be calculated as follows [31]:(3)$$LM=CNR_{link}-CNR_{required}$$
(4)$$CNR_{link}=P_{t}+G_{t}+G_{r}-L_{f}-N_{0}$$
(5)$$CNR_{required}=E_{b}/N_{0}+10\log_{10}B_{r}-G_{c}+G_{d}$$
(6)$$N_{0}=10\log_{10}k+10\log_{10}T_{i}$$
(7)$$T_{i}=T_{0}\left(NF-1\right)$$

The parameters required in the calculation are listed in Table 5. Assuming the proportion of the signal power received by an outside antenna at a specified distance to the noise power density of an implanted antenna emitting at a specified power, the required CNR is the carrier-to-noise ratio needed at the receiving end to satisfy a certain communication rate and BER requirements and is associated with the sensitivity of the receiver [31]. Here, we used BPSK modulation, which requires a BER of less than 1 × 10^−5^ and a bit rate Br of 1 Mb/s [31]. Currently, the input power of the antenna when operating at 1.4 GHz and 2.4 GHz is 4.24 dBm and 5.02 dBm, respectively, and the outer receiving antenna adopts a circularly polarized antenna with a gain of 2.15 dBi. Transceiver ranges of 8.2 m and 9.7 m are achieved at 1.4 GHz and 2.4 GHz, respectively, when the communication link margin exceeds 20 dB (Table 5).

## 6. Conclusions

In this paper, we designed a small dual-band, dual-polarized, implantable antenna for an implantable medical monitoring system. We analyzed the operating principle of the antenna in detail and obtained a good set of antenna parameters. To verify the actual radiation capability of the antenna, a prototype was made and measured in biogel and minced pork. The results of the measurements reveal that the −10 dB bandwidth covers the 1.4 GHz WMTS band as well as the 2.45 GHz ISM band very well, and the radiation patterns have good symmetry. As its size is only 47.4 mm^3^ and it has good radiation characteristics, this antenna is a fine product and candidate for use in implantable medical equipment.

## Figures and Tables

**Figure 1 sensors-23-09094-f001:**
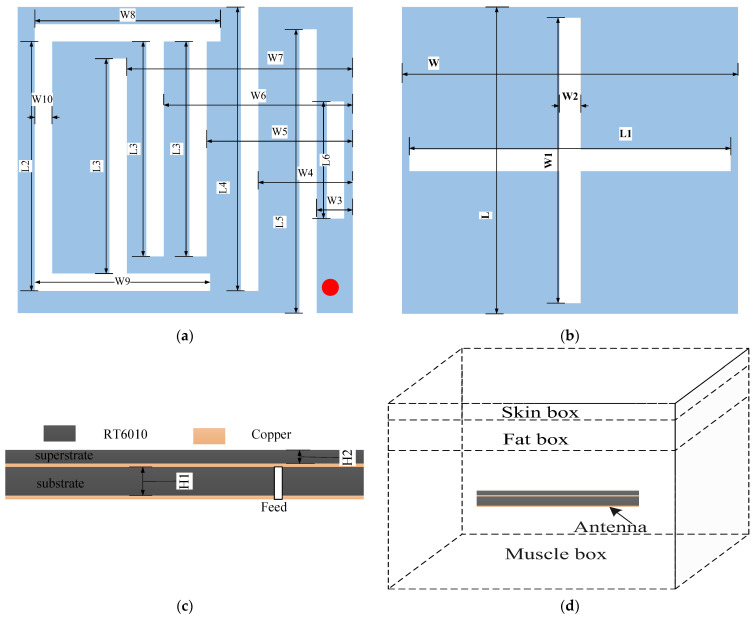
Design of dual-band antenna (not to scale). (**a**) Patch with slots. (**b**) GND with two crossed, rectangular slots. (**c**) Side view of the antenna. (**d**) Antenna within a three-layer body tissue model.

**Figure 2 sensors-23-09094-f002:**
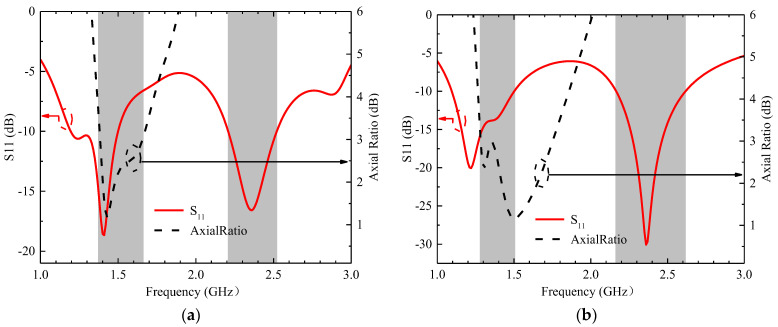
The S11 and AR simulation results at different operating frequencies: (**a**) at 1.4 GHz; (**b**) at 2.4 GHz.

**Figure 3 sensors-23-09094-f003:**
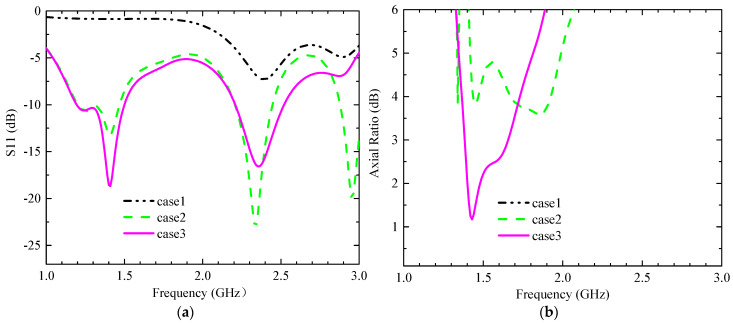
Simulation results of S11 and AR in three cases. (**a**) S11 and (**b**) AR.

**Figure 4 sensors-23-09094-f004:**
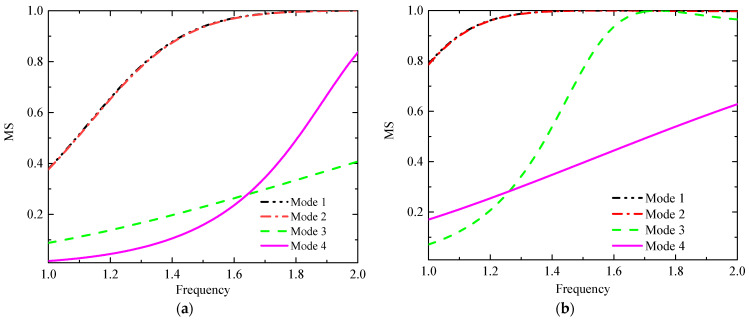
The simulated MS of different cases. (**a**) Case 2 and (**b**) case 3.

**Figure 5 sensors-23-09094-f005:**
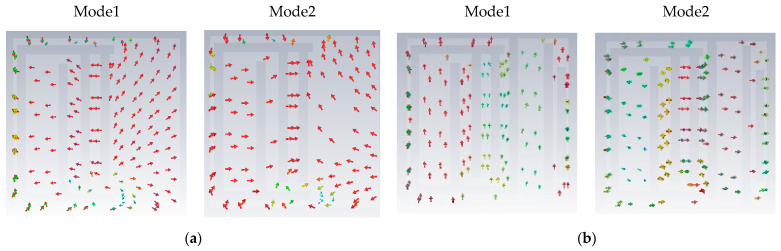
The simulated current vector distributions on the radiation patch of different cases. (**a**) Case 2 and (**b**) case 3. (The darker the arrow color, the greater the current intensity).

**Figure 6 sensors-23-09094-f006:**
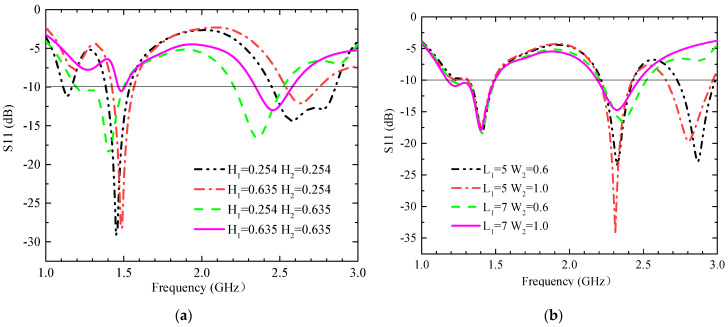
The simulated S11 values when key parameters are changed. (**a**) The thicknesses of the substrate (H_1_) and superstrate (H_2_) at different sizes; (**b**) the length (L_1_) and width (W_2_) of the cross groove at different sizes.

**Figure 7 sensors-23-09094-f007:**
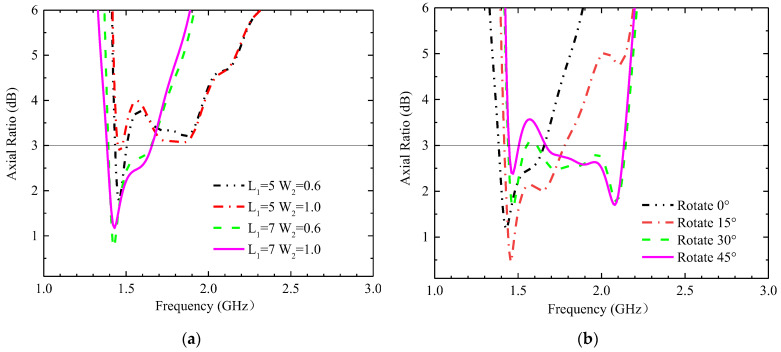
The simulated AR when some key parameters are changed. (**a**) The length (L_1_) and width (W_2_) of the cross-slot at different sizes, and (**b**) the rotation angle of the cross groove.

**Figure 8 sensors-23-09094-f008:**
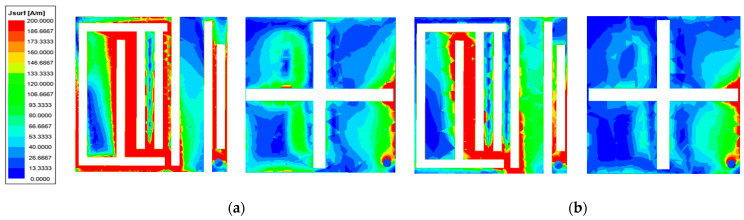
Current distributions on antenna radiation patch and GND: (**a**) 1.4 GHz; (**b**) 2.45 GHz.

**Figure 9 sensors-23-09094-f009:**
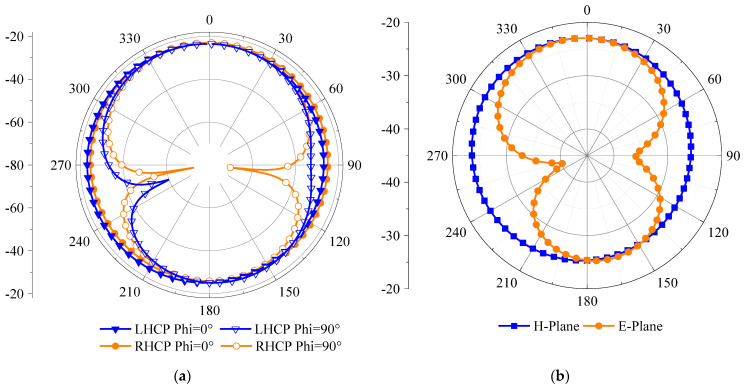
The simulation results of the radiation patterns: (**a**) 1.4 GHz and (**b**) 2.45 GHz.

**Figure 10 sensors-23-09094-f010:**
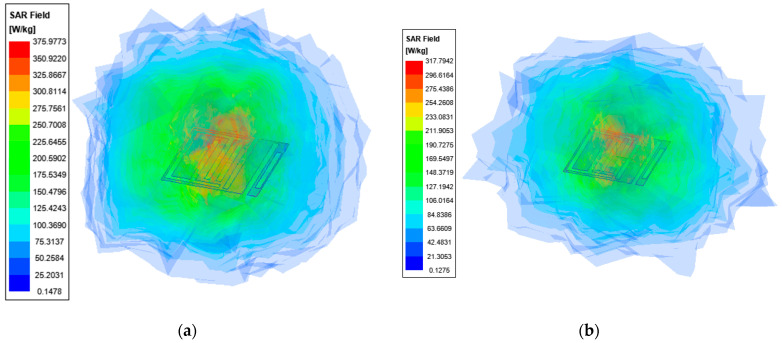
SAR distribution in skin model (1 g) at 1.4 GHz (**a**) and 2.45 GHz (**b**).

**Figure 11 sensors-23-09094-f011:**
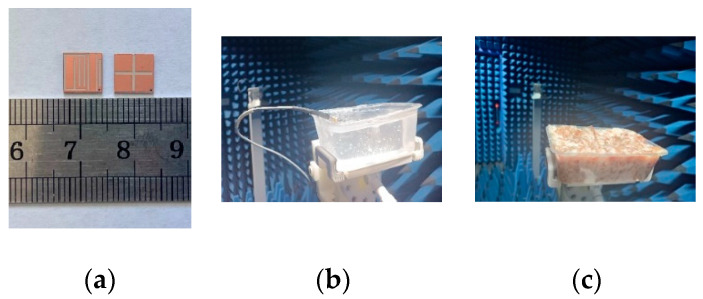
Antenna prototype and experimental overview. (**a**) Prototype, (**b**) the antenna in biogel, and (**c**) the antenna in minced pork.

**Figure 12 sensors-23-09094-f012:**
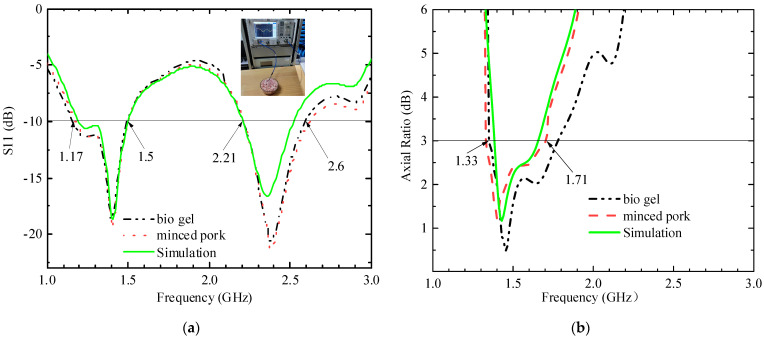
Comparison of simulated and measured S11 and AR values under different conditions. (**a**) S11 and (**b**) AR.

**Figure 13 sensors-23-09094-f013:**
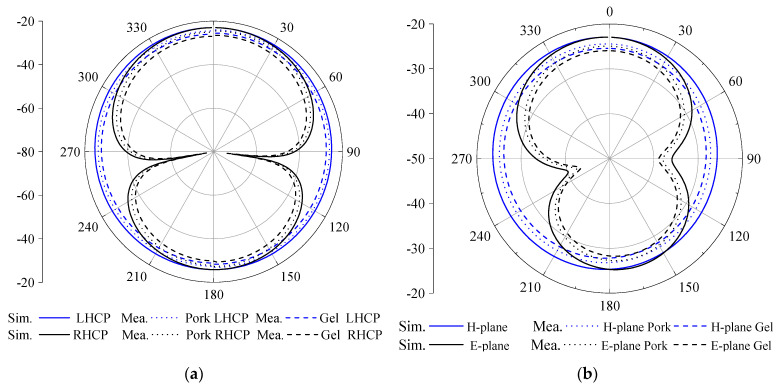
Comparison of the measured and simulated radiation patterns. (**a**) 1.4 GHz; (**b**) 2.45 GHz.

**Table 1 sensors-23-09094-t001:** Comparison of the designed antenna with recent works.

Ref.	Vol. (mm^3^)	Freq. (GHz)	Bandwidth (MHz)	Gain (dBi)	SAR (1-g)	CP
[6]	120.69	2.45	203	−22.7	733.5	Yes
[15]	102.87	104 2.45	150 158	−29.98 −33.07	507 436	No No
[16]	103.7	1.4 2.45	150 540	−32 −31.6	702 781	Yes No
[17]	127	1.4 2.45	50 153	−37 −21	215 565	No No
[19]	154.88	1.4 2.45	45 60.7	−25.7 −39.9	N/A N/A	No No
[20]	614.4	1.4 2.45	37 100	−13.25 −11.3	N/A N/A	No No
[21]	91.4	1.4 2.45	190 370	−29.4 −21.2	500 686	No No
This Work	47.4	1.4 2.45	310 310	−24.6 −23.8	452 358	Yes No

**Table 2 sensors-23-09094-t002:** Sizes of the antenna’s geometric parameters (unit: mm).

Parameter	Value	Parameter	Value	Parameter	Value
*W*	7.2	*L*	7.4	*W* _1_	7
*W* _2_	0.6	*W* _3_	0.7	*W* _4_	2.25
*W* _5_	3.05	*W* _6_	3.9	*W* _7_	4.85
*W* _8_	4.2	*W* _9_	4.1	*W* _10_	0.4
*L* _1_	7	*L* _2_	6	*L* _3_	5.6
*L* _4_	7.1	*L* _5_	7.2	*L* _6_	5
*H* _1_	0.635	*H* _2_	0.254		

**Table 3 sensors-23-09094-t003:** Dielectric constants and conductivities of different body tissues at two typical frequencies.

Tissue\Frequency	1.4 GHz		2.45 GHz	
	*ε* _r_	*σ* (s/m)	*ε* _r_	*σ* (s/m)
Skin	39.66	1.036	38.0	1.464
Fat	5.56	0.041	5.28	0.104
Muscle	54.1	1.14	52.729	1.73

**Table 4 sensors-23-09094-t004:** Simulation of the evolution process of the antenna’s geometry.

Case	(1)	(2)	(3)
Top View	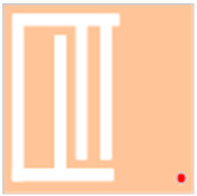	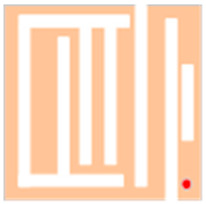	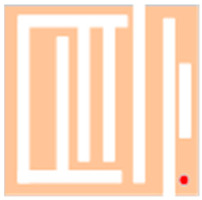
Bottom View	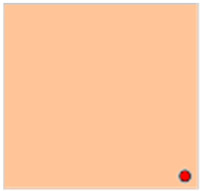	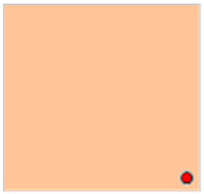	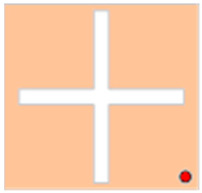

**Table 5 sensors-23-09094-t005:** Parameters of the link budget.

Transmitter		
Operating frequency	1.4 GHz	2.45 GHz
Tx power *P_t_* (dBm)	4.24	5.02
Tx antenna gain *G_t_* (dBi)	−18.8	−19.1
**Receiver**		
Rx antenna Gain *G_r_* (dBi)	2.15	2.15
Polarization	CP	CP
Ambient temperature *T*_0_ (K)	293	293
Receiver noise figure NF (dB)	3.5	3.5
Boltzmann constant *k*	1.38 × 10^−23^	1.38 × 10^−23^
Noise power density (dB/Hz)	−199.95	−199.95
**Signal Quality**		
Bite rate *B_r_* (Mb/s)	1	1
Bite error rate	1.0 × 10^−5^	1.0 × 10^−5^
*E_b_*/*N*_0_ (ideal BPSK) (dB)	9.6	9.6
Coding gain (dB)	0	0
Fixing deterioration *G_d_* (dB)	2.5	2.5

## Data Availability

Data are unavailable due to privacy or ethical restrictions.

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
