# Peer review of "Designing a Compact Dual-Band, Dual-Polarized Antenna for Biotelemetry Communications Using the Characteristic Mode Method"

_sensors, 2023, doi:10.3390/s23229094_

Round 1

Reviewer 1 Report

Comments and Suggestions for Authors

Comments

1. The contribution of this work is very limited. Only achieving compact size is merely a novelty as discussed by authors line no. 75

2. The authors need to discuss in detail about the methodology of achieving dual polarization? What is the purpose of dual polarized antenna for the intended applications as mentioned in this article?

3. Why in comparison table 1, the proposed work is not included?

4. The authors must need to cite and compare published works from 2023, 2022 in table 1

5. The CMA analysis is very short and incomplete, it seems just authors tried to draw the attention so they put it title, what is the relevancy, significance of CMA for this designed antenna?

6. Which type of feeding is used in the fabricated antenna? No feeding is shown

7. Authors need to show clear VNA snapshot (direct display of S11 in VNA by connecting the antenna)

8. Why there are no discrepancy in measured results?

9. The radiation patterns are drawn exactly following envelope of simulation curve. Is it possible to get so much closeness for a tiny antenna. explain the measurement process. Share the data in zip file to validate

10. Why link budget analysis is performed?

11. Explain in detail, how the designed antenna will be suitable for Biotelemetry Communication applications?

12. The authors need to validate the design formula given in equation 1. In fact there are no w and l in antenna geometry

13. why gain, efficiency are not shown?

14. Its not clear why Antennas within three-layer body tissue model is shown?

Comments on the Quality of English Language

Need grammatical corrections 

Author Response

Thank you for your hard work. Your suggestions have been very helpful in improving the quality of the paper. We have made modifications one by one according to your feedback, please refer to the attachment for details.

Reviewer 2 Report

Comments and Suggestions for Authors

In this work, the design of a dual-band dual-polarized compact implantable antenna using the characteristic mode method for wireless medical telemetry service (WMTS) and industrial, scientific, and medical (ISM) bands. This work is of interest, however, there are some concerns of this reviewer to be addressed. Please find below my comments:

1. Can you elaborate on the specific applications and use cases for which this dual-band dual-polarized antenna is intended within the field of biotelemetry communication?

2. What motivated the choice of the characteristic mode method for designing this implantable antenna, and how does it contribute to its performance characteristics?

3. How do the slotting techniques and high dielectric constant materials employed in the design help in minimizing the antenna's volume?

4. Could you explain the significance of achieving circular polarization in the WMTS band and its potential implications in medical telemetry applications?

  5. In the three-layer human tissue simulation, how accurately does the antenna's performance in the lab match what might be expected within the human body?

6.  How do the measured peak gains and SAR values at 1.4 GHz and 2.45 GHz relate to safety standards, and what are the practical implications for implantable devices?

7.  What design considerations were made to ensure that the antenna remains compact while still meeting the stringent requirements for human implantable devices?

8.  Can you provide insights into the challenges and trade-offs encountered when trying to maintain efficiency and gain, especially within the human body's lossy tissues?

9.  How might this compact dual-band dual-polarized antenna enhance the reliability and performance of implantable medical devices, particularly in terms of biocompatibility and lifetime?

10. In the conclusions, you mention that this antenna is a "fine product candidate." What additional research or development steps are needed before it can be applied in real-world implantable medical monitoring systems?

Comments on the Quality of English Language

Minor edits are required.

Author Response

(The authors gave the same response as above.)

Round 2

Reviewer 1 Report

Comments and Suggestions for Authors

The authors have answered all the comments.

Comments on the Quality of English Language

Minor edits still required.

Reviewer 2 Report

Comments and Suggestions for Authors

Thank you for addressing the comments of this reviewer. I have no more comments.

Comments on the Quality of English Language

Acceptable